# Types of Cell Death from a Molecular Perspective

**DOI:** 10.3390/biology12111426

**Published:** 2023-11-13

**Authors:** Fatemeh Hajibabaie, Navid Abedpoor, Parisa Mohamadynejad

**Affiliations:** 1Department of Biology, Faculty of Basic Sciences, Shahrekord Branch, Islamic Azad University, Shahrekord 88137-33395, Iran; fateme.hajibabaii1991@gmail.com; 2Department of Physiology, Medicinal Plants Research Center, Isfahan (Khorasgan) Branch, Islamic Azad University, Isfahan 81551-39998, Iran; 3Biotechnology Research Center, Shahrekord Branch, Islamic Azad University, Shahrekord 88137-33395, Iran; 4Department of Sports Physiology, Faculty of Sports Sciences, Isfahan (Khorasgan) Branch, Islamic Azad University, Isfahan 81551-39998, Iran

**Keywords:** programmed cell death, uncontrolled cell death, molecular pathways, cell morphology

## Abstract

**Simple Summary:**

Inflammation and free radicals can stimulate cell self-destruction. Inflammation and cell death are vital aspects of most diseases. Accumulation of cell damage leads to the impairment and dysregulation of the cell function. Thus, understanding the pathomechanism and molecular signaling pathways involved in cell death is necessary. Moreover, recognizing the factors that stimulate cell death can help in providing insights for formulating a new strategy for comprehending the management and treatment of cell function.

**Abstract:**

The former conventional belief was that cell death resulted from either apoptosis or necrosis; however, in recent years, different pathways through which a cell can undergo cell death have been discovered. Various types of cell death are distinguished by specific morphological alterations in the cell’s structure, coupled with numerous biological activation processes. Various diseases, such as cancers, can occur due to the accumulation of damaged cells in the body caused by the dysregulation and failure of cell death. Thus, comprehending these cell death pathways is crucial for formulating effective therapeutic strategies. We focused on providing a comprehensive overview of the existing literature pertaining to various forms of cell death, encompassing apoptosis, anoikis, pyroptosis, NETosis, ferroptosis, autophagy, entosis, methuosis, paraptosis, mitoptosis, parthanatos, necroptosis, and necrosis.

## 1. Introduction

Over the course of the past ten years, the nomenclature committee on cell death has dedicated its efforts to the establishment of criteria for the systematic classification and analysis of cell death, encompassing morphological, biochemical, and functional aspects [1,2]. Srinivasan et al. presented a comprehensive analysis of the advancements achieved by computational and systems biologists in elucidating the many regulatory mechanisms involved in cell death. These mechanisms together form the intricate network responsible for controlling cell death processes. The cell death network is characterized as an all-encompassing decision-making process that regulates many biochemical circuits responsible for executing cell death [3]. This network incorporates a variety of feedback and feed-forward loops, as well as the crosstalk across several pathways involved in the regulation of cell death. Indeed, comprehending the intricate dynamics of these intricate regulatory processes necessitates the use of mathematical modeling and system-oriented methodologies [3]. Mathematical modeling functions as a potent instrument for establishing a connection between molecular biology and cell physiology. It achieves this by linking the qualitative and quantitative characteristics of dynamic molecular networks with signal-response curves that are recorded by cell biologists [4]. The dynamics of complex molecular networks that regulate the cell cycle [5,6], nutritional signaling [7], checkpoints [8], signaling dysregulation in cancer [9], and cell death [10,11,12,13] have been effectively described using mathematical and systems-oriented methodologies. Without a doubt, the control of cell death is a molecular process that needs mathematical modeling in order to attain a comprehensive understanding of the process. In a cell’s lifetime, there are four main biological processes: survival, cell division, differentiation, and cell death [2]. Eliminating damaged cells and maintaining the organism’s homeostasis are two of the primary functions of cell death during embryonic development [14]. A cell dies when it stops dividing and functioning as a part of a living organism. This phenomenon may occur because of the body’s normal cellular turnover rate, as a consequence of disorders or localized damage, or as a result of the organism’s death, from which the cells originate [15,16]. There were initially three types of cell death [1] (Figure 1):Type I cell death (apoptosis);Type II cell death (autophagy);Type III cell death (necrosis).

Although regulated cell death (RCD) is commonly associated with maintaining organismal homeostasis in both physiological and pathological contexts, it is worth noting that RCD is not exclusive to multicellular organisms. Unicellular eukaryotes, such as *Dictyostelium discoideum,* and prokaryotic organisms, such as *Escherichia coli*, also exhibit regulated cell death [17,18]. Moreover, there is instantaneous and catastrophic cell death in contrast to regulated cell death. The cell death category occurs by an exposure to severe physical, chemical, or mechanical attacks [19]. It is worth noting that RCD requires a specialized regulatory network, implying that it can be modulated (accelerated or delayed) by pharmacological intermediation or genetic modification. RCD is implicated in two very different scenarios, despite the underlying molecular pathways showing substantial similarities [20]. RCD may occur without any direct environmental disruption, acting as an established trigger of physiological systems for proliferation or tissue regeneration. RCD may originate from extracellular or intracellular microenvironmental effectors and is disturbed in a way considered as an acute adaptive mechanism for homeostasis offset and stress suppression.

Adaptive stress responses are similar to stress-driven RCD since both aim to maintain a state of biological homeostasis. Despite cellular homeostasis, which controls adaptative stress responses at the cellular level, RCD contributes directly to organism or colony levels. This homeostatic mechanism not only involves the removal of dysfunctional or harmful cells but also allows for the release of chemicals component from dying cells that serve as an early warning system for the other neighboring cells. Common names for these warning signals include damage-associated molecular patterns (DAMPs), pathogen-associated molecular patterns (PAMPs), and alarmins [21,22,23,24]. 

Cell death strategies may be divided into two groups, programmed and non-programmed cell death, depending on whether they rely on a signal to initiate death [17]. Intracellular signal transduction pathways are the systems that commit cells to programmed cell death (PCD) mechanisms [25]. PCD can be subdivided into non-apoptotic and apoptotic cell death based on morphological properties and molecular interactions [26]. The membrane of the dying cell is conserved during the caspase-dependent process of apoptosis. However, evidence shows that caspases trigger the activation of gasdermin proteins whose N-terminus fragments create pores within the membrane which, upon accumulation, end up causing plasma membrane rupture. For example, caspase-dependent cell death (such as pyroptosis) can cause membrane rupturing as an exception to this classification. In contrast, in scientific terms and based on previous studies, caspase-independent cell death and membrane rupturing are considered non-apoptotic cell death signs [14,25]. 

When it comes to maintaining a homoeostatic balance in multicellular organisms, the body of an organism continuously attempts to keep the number of new cells formed during mitosis equal to the number of damaged or unnecessarily destroyed cells [27]. Large numbers of regulatory genes are required for controlling cell cycling processes that identify cellular abnormalities and trigger apoptosis, a kind of programmed cell death [14]. Many of these regulatory genes either promote or suppress mitosis, as well as begin apoptosis, autophagy, pyroptosis, and another type of programmed cell death. Diseases like cancer, which may spread throughout an organism and eventually kill it, result from uncontrolled cell division [28]. In contrast, degenerative statuses such as rheumatoid arthritis, Parkinson’s, and Alzheimer’s result from excessive cell death rates [29,30]. In light of the extensive and complex interplay of RNAs and proteins inside the cellular processes of the cell cycle and cell death, some regulatory proteins, receptors, and enzymes have been identified as key regulators. Mutations or aberrant expression of these regulators may directly affect the cell cycle machinery [31,32].

Distinct macroscopic morphological changes accompany the death of cells. The utilization of morphotypes has been employed to categorize cell death into three distinct categories, predicated with the methodologies employed for the elimination of diseased cells and their fragments [33]. 

## 2. Types of Cell Death

### 2.1. Apoptosis

Apoptosis or Type I cell death is associated with the following cellular events:Cytoplasmic shrinkage;The irreversible condensation of chromatin in the nucleus (pyknosis);The destructive fragmentation of the nucleus (karyorrhexis);The formation of apoptotic bodies based on the establishment of intact small vesicles;The phagocytosis and decomposition of apoptotic bodies in neighboring cells’ lysosomes [34].

In a publication in 1972, Kerr, Wyllie, and Currie proposed the word “Apoptosis” to characterize a specific form of cell death [35]. Initiating the apoptosis process is associated with the stopping of the proliferation and division of the cell. In contrast, the cellular entity undergoes a regulated mechanism resulting in its demise, while the intracellular contents remain contained within the confines of the cellular milieu. Apoptosis is recognized as a cellular suicide process that is established by triggering a series of cysteine-aspartic proteases that is term as the caspases activation cascade. Caspases may be divided into two classes: those that act as “Initiators Caspase” and those that act as “Executioners Caspase” [36] (Figure 2). 

At the time cell damage is sensed, initiator procaspases 8 and 9 are converted into active initiator caspases, and consequently, they induce the executioner caspases’ activation (caspases 3, 6, and 7). An array of processes occur in apoptotic bodies’ production and destruction of damaged cells, including DNA fragmentation, telomeres’ shortening, the degradation of proteins/cytoskeleton/crosslinking of a protein, phagocytic cell ligands’ expression, and apoptotic body formation [37]. As a genetically conserved mechanism, apoptosis is highly regulated in multicellular organisms. There are two pathways for the apoptosis process:The intrinsic pathway based on intracellular damage sensors’ detection.The extrinsic pathway based on immune cell and damaged cell attachment.

The apoptotic cell death rate in humans is around 1 × 10^9^ cells/day [38].

The intrinsic pathway of apoptosis

The intrinsic route of apoptosis, also referred to as the mitochondrial pathway, involves the interaction of many stimuli with diverse cellular targets. [39]. This kind of apoptosis is triggered by either a positive and/or negative pathway and relies on substances produced by the mitochondria [40]. A deficiency of growth factors, cytokines, and hormones in the cellular microenvironment might trigger a negative signal to initiate the apoptosis process [14]. Apoptosis is triggered by activating pro-apoptotic molecules like p53 upregulated modulator of apoptosis (Puma), Noxa, and Bcl-2-associated X (Bax) without pro-survival signals [41]. Apoptosis may also be triggered by an exposure to positive variables such as hypoxia, poisons, radiation, reactive oxygen species (ROS), viruses, and other hazardous agents, yet in the case of certain cells, like neutrophils, hypoxia can enhance cell survival [42]. Mitochondrial outer membrane permeabilization (MOMP) is a pivotal stage in this process. The modulation of MOMP is influenced by the activity of several pro-apoptotic and anti-apoptotic members belonging to the BCL2 protein family, which serves as an apoptosis regulator. Upon exposure to an apoptotic stimuli, the mitochondrial outer membrane permeabilization process is triggered, resulting in the sequential activation of the initiator caspase 9 (CASP9) followed by the executioner caspases CASP3 and CASP7 [43]. Researchers have successfully discovered two separate classes of pro-apoptotic BCL2 proteins that exhibit different functional characteristics. The first category comprises the apoptotic activators BAX, BAK1, and BOK. Upon being triggered by apoptotic signals, BAX, BAK1, and BOK initiate MOMP by creating holes in the outer mitochondrial membrane (OMM). These pro-apoptotic factors stimulate the release of many apoptogenic substances such as Cytochrome-C and diablo IAP-binding mitochondrial protein (DIABLO; also known as second mitochondrial activator of caspases, SMAC) into the cytoplasm. The apoptogenic activity of Cytochrome-C is manifested by its interaction with apoptotic peptidase activating factor 1 (APAF1) and pro-CASP9, resulting in the formation of a complex referred to as the apoptosome. This complex then triggers the sequential activation of CASP9, as well as of the executioner caspases CASP3 and CASP7. The activation of CASP3 and CASP7 is facilitated by the interaction of DIABLO/SMAC with X-linked inhibitor of apoptosis (XIAP) and other members that are inhibitors of the apoptosis (IAP) protein family [43]. The second category of pro-apoptotic BCL2 proteins, referred to as BH3-only proteins, encompasses a group of molecules including BAD, PUMA, BIK, BIM, BMF, BID, HRK, and NOXA. Direct interaction between caspase-cleaved BID (tBID), BIM, PUMA, and NOXA in the mitochondria has the capability to facilitate the activation of BAX and BAK1. On the other hand, indirect activation of BAX and BAK1 occurs when BH3-only proteins including BAD, BIK, BMF, and HRK bind to and inhibit the activity of anti-apoptotic BCL2 family members [44,45]. Caspase-9 is the initiator caspase that regulates the intrinsic mechanism of apoptosis by binding to the apoptotic protease activating factor-1 (APAF1) once its caspase recruitment domains (CARD) have been exposed [14,46]. In nonactive apoptotic cells, APAF1 is often folded to prevent its CARD domain from binding to procaspase-9 [47]. The interaction between Cytochrome-C and the tryptophan-aspartic acid (WD) domain of APAF1 monomers induces a structural change in APAF1, leading to the exposure of a region responsible for nucleotide binding and oligomerization. This region specifically binds to deoxyadenosine triphosphate (dATP), hence triggering the initiation of apoptosis [48]. Due to the extra conformational shift induced by this interaction, the CARD and oligomerization domains of APAF1 are exposed, allowing for the assembly of several APAF1s into an apoptosome [49]. Many procaspase-9 proteins are recruited and activated by the apoptosome’s exposed CARD domains, which are located in the open core of the cell death complex [14]. When the caspase 9 is activated, it triggers the executioner procasp-3, which, once converted to active caspase-3, causes the complete induction of apoptosis. However, gasdermin E, a substrate of active caspase-3, induces pyroptosis, rather than apoptosis. While apoptosis may be triggered by the actions of Smac/Diablo and HtrA2/Omi, the inhibition of inhibitors of apoptosis proteins (IAPs) is inadequate without the release of Cytochrome-C [50,51]. 

Extrinsic pathway of apoptosis

Extrinsic apoptosis, also called the death receptor (DR) pathway, is triggered when death ligands are released by patrolling NK-cells or macrophages and bind with DRs on the target cell membrane [52]. This triggers the extrinsic route, which in turn activates caspase 8 from pro-caspase-8. The DRs are proteins with structural and functional similarities with the tumor necrosis factor (TNF) superfamily [53,54]. When a death ligand binds to a DR, the DR’s cytoplasmic domain becomes a death-inducting signaling complex (DISC), where monomeric pro-caspase-8 is recruited through its death effector domain (DED) [55,56]. The adaptor protein known as the TNFR-associated death domain (TRADD), or the FAS-associated death domain (FADD), is also part of the DISC. It aids in the binding of pro-caspase-8 [57]. Multiple pro-caspase-8 monomers are recruited to the DISC, where they undergo dimerization and trigger to become caspase-8. Caspase-8 may initiate an apoptosis mechanism by any of two sub-pathways [14,58].

Whether the cells are type I or type II determines which sub-pathway gets activated [59]. Caspase-8 directly cleaves executioner caspases, triggering apoptosis in type I cells [60]. Unless blocked by proteins secreted from the mitochondria, IAPs prevent the direct activation of the executioner caspases by caspase-8. Mice lacking caspase-8, who normally respond to DR ligands, reveal the crucial function that caspase-8 plays in the regulation of the apoptosis extrinsic cascade [51]. Whether apoptosis is initiated by the intrinsic or extrinsic routes, it must be tightly regulated to avoid the disastrous outcomes that might result from insufficient control. Mutations in the multiple apoptosis initiation systems are a common cause of cancer [61,62]. The creation of a benign tumor or cancer results from this phenomenon when it happens in conjunction with a failure to react to external cues that would ordinarily trigger the extrinsic route or prevent proliferation [63].

### 2.2. Anoikis

For the first time, Frisch described the “Anoikis” concept in 1994 [64]. The loss of integrin-dependent anchoring is considered to be the trigger for the anoikis subtype of intrinsic apoptosis [65]. Anoikis, a Greek word meaning “homelessness” or “loss of home,” describes the one type of apoptosis that occurs when cells lose their connection to the extracellular matrix (ECM) and adhere to an unsuitable site [66]. Integrin receptors are mediators of ECM interaction and are essential for migration, proliferation, and survival because they not only establish physical linkages with the cytoskeleton but also transduce signals from the ECM to the cell [67]. By preventing cells from detaching and re-adhering to inappropriate matrices, as well as by inhibiting dysplastic development, anoikis serves as a vital defensive mechanism for an organism [68,69]. Because of this, adherent cells may be able to survive in a suspension or proliferate in ectopic places where the ECM proteins are different if the anoikis program is not well executed [70]. Emerging evidence suggests that cancer cells’ aberrant execution of anoikis is a characteristic of the disease that promotes metastasis to distant organs [65]. Anoikis is a kind of apoptosis that is produced by insufficient or incorrect ECM connections but otherwise follows the same mechanisms as apoptosis [69,71].

The induction and execution of anoikis include many routes that converge on the activation of caspases and subsequent molecular processes. This activation triggers the activation of endonucleases, resulting in DNA fragmentation and ultimately leading to cell death. [72]. Two apoptotic routes, the intrinsic pathway involving mitochondrial dysfunction and the extrinsic pathway involving the activation of cell surface death receptors, work together to initiate the anoikis program (the extrinsic pathway) [65]. Proteins belonging to the B cell lymphoma-2 (Bcl-2) family play important roles in both of these processes. Three subfamilies exist within the Bcl-2 family, and they are as follows [66]:A.Myeloid cell leukemia sequence 1, as well as the anti-apoptotic proteins Bcl-2 and B-cell lymphoma-extra large, Bcl-XL (Mcl-1).B.Pro-apoptotic proteins Bax, Bcl-2 homologous antagonist/killer (Bak), and Bcl-2 related ovarian killer (Bok), all with several domains.C.BH3 interacting domain death agonist (Bid), BCL2-associated agonist of cell death (Bad), Bcl-2 interacting mediator of cell death (Bim), BCL-2 interacting killer (Bik), BCL-2 modifying factor (Bmf), Noxa, Puma, and Harakiri (Hrk) are all pro-apoptotic BH3-only proteins [66].
The intrinsic pathway of Anoikis


DNA damage and endoplasmic reticulum stress are two intracellular cues that initiate the intrinsic pathway of apoptosis. In this process, mitochondria play a crucial role in regulating apoptosis [73]. In response to death signals, the pro-apoptotic proteins Bax and Bak undergo translocation from the cytosol to the outer mitochondrial membrane (OMM). The oligomerization of these proteins leads to the formation of a channel through the OMM, which in turn causes mitochondrial permeabilization. This permeabilization event subsequently triggers the release of Cytochrome-C [74]. In addition to the Bax proteins’ inherent pore-generating activity, their interaction with mitochondrial channel proteins, such as the voltage-dependent anion channels, may also contribute to membrane permeabilization [65]. The activation of the effector caspase-3 occurs subsequent to the release of Cytochrome-C. Apoptosis is initiated by assembling the apoptosome complex, which then triggers the activation of caspase-9 and the cofactor APAF [75]. The BH3-only pro-apoptotic proteins are known to have significant functions in the intrinsic pathway of the anoikis cell death mechanism. Bid and Bim are proteins from this biological family that become active upon cellular detachment from the ECM, hence enhancing the production of Bax-Bak oligomers within the OMM [76]. “Activators” refers to this class of BH3-only proteins [77]. Specifically, Bim is confined to the dynein cytoskeletal complexes until cell separation triggers its release and translocation to the mitochondria [78,79].

The process of Bim phosphorylation by extracellular signal-regulated kinase and phosphatidylinositol 3-kinase (PI3K)/protein kinase B (Akt) is initiated during integrin contact. This phosphorylation event serves to impede the proteasomal degradation of Bim, resulting in its accumulation subsequent to the loss of cell adhesion [80]. Bik, Bmf, Bad, Puma, Hrk, and Noxa are all examples of so-called “Sensitizers”, another class of BH3-only proteins [65]. The inhibitory impact of Bcl-2 on apoptosis is impeded when sensitizer BH3-only proteins engage in a competition for its BH3 binding domain. This competition enables activator BH3-only proteins to initiate the development of Bax-Bak oligomers, hence facilitating cellular demise [81]. The Bmf functions as a surveillance mechanism in epithelial cells, detecting disruptions in the cytoskeleton structure and transmitting signals that initiate cell death [82]. Following the process of cell separation, Bmf becomes dissociated from the myosin V motor complex and subsequently accumulates within the mitochondria. Within this organelle, Bmf acts to counteract the effects of Bcl-2, thus initiating the release of Cytochrome-C and ultimately leading to the execution of anoikis [65].

The extrinsic pathway of Anoikis

The execution of anoikis involves the activation of both the intrinsic and extrinsic pathways. The initiation of the DISC occurs in the extrinsic route when a ligand binds to death receptors belonging to the TNFR superfamily. These death receptors include the Fas receptor, TNF receptor superfamily 1 (TNFR1) receptor, and the TNF-related apoptosis-inducing ligand (TRAIL) receptors-1 and TRAIL receptors-2. DISC, by means of engaging with adaptor proteins such as FADD, facilitates the aggregation of numerous caspase-8 molecules and triggers their subsequent activation [83,84]. Substrate proteolysis and cell death result from the active caspase-8 being secreted into the cytoplasm, where it cleaves and activates the effector caspases, caspases-3, caspases-6, and caspases-7 [77,85]. An alternative mechanism that connects the extrinsic and intrinsic pathways is the cleavage and activation of Bid upon Caspase-8 activation [86,87]. This t-Bid version can induce mitochondrial Cytochrome-C release and apoptosome assembly. Detachment of cells from the ECM has been shown to trigger the release of a mitochondrial protein called Bit1 into the cytoplasm, where it functions as a pro-apoptotic mediator and induces a caspase-independent type of apoptosis [65]. Mitochondrial damage in certain cases is a subsequent effect of the occurrence of death receptor activation, which may be created as a feedback loop between extrinsic death signals and the intrinsic route. Prior studies have demonstrated the significance of the extrinsic pathway in the occurrence of anoikis, wherein the detachment from the ECM triggers the increased expression of Fas and Fas ligand, while simultaneously reducing the levels of FADD-like interleukin-1β-converting enzyme-like inhibitory protein (FLIP), an inherent inhibitor of Fas-mediated signaling [88]. Morphological alterations in cells are another interesting trigger for the extrinsic apoptosis pathway. The rounded shape of a cell, after its detachment has occurred, may cause “induced proximity” of Fas receptors, triggering their activation [65]. Cell death occurs through the convergence of both the extrinsic and intrinsic apoptotic pathways, which are dependent on the activation of the effector caspase-3. The activation of caspase-3 triggers a subsequent proteolytic cascade and exerts an influence on the cellular apoptosis pathway. The cleavage of signaling molecules, such as focal adhesion kinase (FAK) and protein 130 kDa Crk-associated substrate (p130Cas), is of utmost importance for the effective implementation of apoptosis [89,90]. When FAK is cleaved by caspases, it interferes with the structure of focal adhesions and dampens the survival signal they provide. A caspase-mediated cleavage disrupts the localization and interactions of p130Cas with paxillin as an SH2/SH3 adaptor protein that binds to FAK and transmits integrin signals [91]. On the other hand, the C-terminal inhibitory fragment generated by p130Cas cleavage hinders the transcription of p21^Waf1/Cip1^. Hence, this inhibitory fragment triggers an apoptotic response and also a cell cycle arrest [65].

Death-associated protein 3 (DAP3) is an important regulator of both types of programmed cell death, apoptosis and anoikis. The most common form of PCD, apoptosis, may occur in response to DNA damage that cannot be repaired or after being induced by inflammatory cells. When epithelial cells are detached from their extracellular matrix, they undergo a kind of apoptosis called anoikis. Integrins and the extracellular matrix (ECM) have different requirements for the various cell types. Internal signaling of these connections into the apoptosis pathways is also being defined, and it is possible that it will differ across cell types. The common execution route is thought to be the final destination for these signals after they have been sent via the internal and external apoptotic pathways [68]. In this part, the difference between anoikis and apoptosis cell death based on the definition, induction, and function in Table 1.

### 2.3. Pyroptosis

Pyroptosis cell death is a specialized form of RCD that is often formed as a result of inflammatory caspase activation and is dependent on the creation of plasma membrane pores by members of the gasdermin protein family. Pyroptosis refers to an inflammatory mode of cellular demise that is initiated by intracellular sensors, such as NLRP3, which are capable of detecting various triggers including damage-associated molecular patterns (DAMPs), pathogen-associated molecular patterns (PAMPs), disruptions in cellular membranes, imbalances in osmotic conditions, and the efflux of ions. Upon initiation, the aforementioned sensors engage the adapter apoptosis-associated speck-like protein containing a CARD (ASC), resulting in the formation of a micron-scale complex known as the inflammasome. The oligomeric aggregates serve as platforms facilitating the activation of caspase-1. Caspase-1, in its active state, performs the cleavage of proforms of the cytokines IL-1β and IL-18, which belong to the interleukin family. Caspase-1 is responsible for the activation of gasdermin D (GSDMD), which results in the exposure of the N-terminal domain. This exposed domain is responsible for the formation of holes in the plasma membrane, facilitating the subsequent release of mature IL-1β and IL-18. This process leads to cell swelling, characterized by a “ballooning effect”, and ultimately triggers pyroptosis. Moreover, the detection of lipopolysaccharide (LPS) inside cells may lead to a process known as pyroptosis. This occurs when guanylate-binding proteins (GBPs) attach to the outer surface of bacteria and form a complex that activates caspase-4/11. This complex consists of GBP1, GBP2, GBP3, and GBP4, as well as the cytosolic forms of caspase-4 and caspase-11. The process of pyroptosis is initiated by the enzymatic activity of active caspase-4 and -11, which results in the cleavage of gasdermin D (GSDMD). It is worth mentioning that caspase-4 has the ability to cause proteolytic cleavage of IL-18. Pyroptosis, being an intrinsically inflammatory mode of cellular demise, has many tiers of regulatory mechanisms. The upregulation of NLRP3 and the production of cytokine IL-1 need initial activation via the stimulation of Toll-like receptors (TLRs), tumor necrosis factor receptors (TNFRs), or IL-1 receptors (IL-1Rs), which then leads to the activation of NFκB. Posttranslational modifications, including phosphorylation and ubiquitylation, play a crucial role in the regulation of inflammasome sensors, as seen in the above diagram. Additionally, the process of cleavage also contributes to the regulation of these sensors [92]. Evidence is mounting that suggests the pore-forming and pyroptotic activities of the N-terminal domains of gasdermins such GSDMA, GSDMB, GSDMC, GSDME/DFNA5, and GSDMA3 are similar to those of GSDMD. Inflammatory caspase activation, which often happens following the identification of intracellular pathogens in immune cells, results in pyroptosis or capase-1-dependent cell death [93]. Both intrinsic and extrinsic apoptosis are considered “classic” because they include the separation of the cell’s internal components and the elimination of damaged cells without harming surrounding cells. It has been discovered that there is a kind of apoptosis that, although it is still regulated by a caspase-dependent set of events, promotes inflammation (proinflammatory events) [14,93]. As opposed to caspase-1-deficient cells, macrophages infected with Salmonella or Shigella undergo a type of cell death known as pyroptosis.

The activation of Caspase-1 by pathogens leads to cellular demise, accompanied by the liberation of inflammatory cytokines into the surrounding milieu. This occurs through the processing of the precursor forms of IL-1β and IL-18, converting them into their bioactive states [94]. As a consequence of pro-caspase-1 being cleaved into active caspase-1, pores emerge in the plasma membrane of the damaged cell [95]. When water enters the cell via the pores, edema and lysis occur due to the intracellular and extracellular ionic gradients not having significant differences. When a cell undergoes pyroptosis instead of apoptosis, the nucleus is conserved (without fragmentation), despite the fact that the nucleus condenses. Evidence from the central nervous system and the cardiovascular system indicates that pyroptosis is a physiologically relevant type of cell death [14,96] (Figure 3). On the other hand, inflammation is often seen in cases of necrotic death [1]. HMGB1 and hepatoma-derived growth factor (HDGF) are two of the factors released by necrotic cells. The nod-like receptor protein 3 (NLRP3), the main protein of an inflammasome, is responsible for sensing HMGB1 and HDGF. Consequently, this triggers inflammasome activation, which in turn releases the pro-inflammatory cytokine IL-1β. When cells are injured, they release ATP, which is then used to activate the NLRP3 inflammasome [97]. The concept of cell death pathways operating independently and with little overlap has been widely accepted for a significant period of time. Currently, it is evident that there exists a strong interconnection between apoptosis, necroptosis, and pyroptosis, leading to a reciprocal regulation among these processes [92]. The role of caspase-8 in facilitating both apoptotic and necroptotic pathways was among the first connections identified between various forms of cellular death [98]. Caspase-8 has a dual role in cellular processes, as it not only governs the mechanism of programmed cell death known as apoptosis but also serves as a pivotal constituent of the ripoptosome. Within the ripoptosome, caspase-8 functions as a critical mediator for the cleavage of RIPK1 and RIPK3, and acts as one of the enzymes responsible for the deubiquitylation of TRAF2 or RIPK1, namely, CYLD [99]. Caspase-8 functions to inhibit the assembly of necrosomes, hence promoting the occurrence of apoptosis in preference to necroptosis. The activation of caspase-8 by FADD after the activation of death receptors initiates apoptosis, while the lack or pharmacological inhibition of caspase-8 leads to necroptosis [92,100]. It is noteworthy that caspase-8 seems to have a function in the stability of the ripoptosome, although its proteolytic activity is necessary to inhibit necroptosis [101]. Multiple studies have provided evidence that RIPK3 has a regulatory role in the activity of caspase-8 and the consequent activation of NLRP3, which occurs downstream of Toll-like receptor (TLR) or tumor necrosis factor receptor 1/2 (TNFR1/2) [102,103]. Recent studies indicate that caspase-8 facilitates the activation of NLRP3 by directly cleaving GSDMD. Consequently, this process enhances pyroptosis downstream of TNF and defends against infection, such as Yersinia infection, in vivo [104,105,106]. Remarkably, the activation of caspase-8, which occurs as a result of the intrinsic apoptosis pathway involving mitochondrial instability mediated by BAX/BAK, followed by the activation of caspase-3/7, has recently been linked to the activation of NLRP3 and the subsequent release of bioactive IL-1β from macrophages [107]. Tsuchiya has discovered an additional association between pyroptosis and apoptosis [108]. In the context of macrophages, when GSDMD is not present, the activation of caspase-1 alters the cellular outcome towards apoptosis that is reliant on caspase-3, caspase-9, and Bid. The aforementioned discovery may provide an explanation for the limited decrease seen in macrophage mortality in the absence of Gsdmd, as reported by Kayagaki et al. [109]. The mechanism by which Caspase-1 initiates the activation of Caspase-3 and subsequent death is yet to be fully elucidated. Another example of an immune response that establishes a connection between several cell death pathways is the induction of NLRP3 activation and the subsequent release of IL-1β by RNA viruses. This process occurs in a manner that is reliant on RIPK1 and RIPK3, but independent of MLKL [110]. The phosphorylation of DRP1 by RIPK1/RIPK3 is initiated by a viral infection, leading to mitochondrial damage and the activation of NLRP3, perhaps due to the generation of mitochondrial ROS [111,112]. Oxidative stress is a critical factor in several physiological and pathological processes [113]. The maintenance of cellular homeostasis requires a certain threshold of ROS [113]. The buildup of ROS might elicit a dual impact, primarily influenced by the ROS concentration, cell origin, and activation of cellular signaling [113,114]. An optimal amount of ROS may induce cellular harm, genetic alterations, and inflammatory responses, thus facilitating the genesis and advancement of tumors. Conversely, an excessive buildup of ROS in cancerous cells can trigger cell death via several mechanisms, including apoptosis, necrosis, and autophagy [113]. Research has shown that levels of ROS tend to be elevated in cancer cells compared to their counterparts in normal cells [113]. In light of these conditions, it is seen that cancer cells exhibit an increased susceptibility to assault by an abundance of ROS originating from an external source [115,116]. The important function of mitogen-activated protein kinases (Erk, p38, and JNK) in apoptotic signaling mediated by ROS has been well established [117]. Numerous studies have shown that ROS play a crucial role in orchestrating cellular apoptosis via the modulation of the JNK and p38 MAPK signaling pathways [114,118]. Hence, the concentration of ROS, cell origin, the extent of cellular damage, the type of damage, and the activation of cellular signaling can determine the type of cell death. The induction of pyroptosis has been observed in response to the production of reactive oxygen species (ROS) [119]. The signal transduction pathway involving reactive oxygen species (ROS) and c-Jun N-terminal kinase (JNK) was activated by lobaplatin treatment, leading to the initiation of gasdermin E (GSDME)-mediated pyroptosis in cells of colon cancer origin [120]. The induction of caspase-1-mediated pyroptosis in non-small cell lung cancer (NSCLC) by Polyphyllin VI (PPVI) was seen via the signaling axis of ROS, nuclear factor kappa B (NF-κB), nod-like receptor family pyrin domain containing 3 (NLRP3), and gasdermin D (GSDMD). These findings indicate that PPVI shows promise as a novel therapeutic agent for the treatment of NSCLC [121]. Based on the information presented in this discussion, it is possible that the activation of NLRP3 triggered by RNA viruses may occur as a consequence of caspase-8 involvement during mitochondrial instability, leading to the subsequent activation of caspase3/7 [92]. This theory merits more investigation.

In this part, the difference between pyroptosis and apoptotic cell death is presented based on morphological changes, molecular mechanism, regulation, and mitochondrial participation in Table 2.

### 2.4. NETosis: Neutrophil Extracellular Trap-Associated Cell Death

Neutrophil extracellular traps (NETs) are cytoplasmic granular proteins that are involved in a contentious form of regulated cell death called “NETosis”. The idea of NETosis is characterized by the extrusion of a meshwork composed of fibers carrying chromatin and histones, which are associated with these proteins [122,123]. The available evidence indicates that the extrusion of neutrophil extracellular traps is an essential process in the regulated cell death known as NETosis. This form of cell death is primarily observed in cells coming from the hematological system and is characterized by a mechanism that is dependent on ROS. NETs are formed in response to various microbial and aseptic stimuli, as well as through the activation of certain receptors such as Toll-like receptors (TLRs). These stable extracellular networks serve the purpose of trapping and eliminating germs [124,125].

Ample studies have shown that mitochondrial, rather than nuclear, DNA is a major component of NETs. In addition to their antibacterial properties, NETs have been linked to the development of diseases in humans such as diabetes and cancer [126]. Mast cells, eosinophils, and basophils, in addition to neutrophils, are capable of releasing NET-like structures. It is important to note that NET extrusion is not always followed by cellular lysis. The occurrence of NETotic cell death has been postulated to arise via a signaling cascade involving Raf-1 proto-oncogene, serine/threonine kinase (Raf-1), mitogen-activated protein kinase kinase (MAP2Ks), and extracellular signal-regulated protein kinase 2 (ERK2). This cascade ultimately leads to the activation of NADPH oxidase and the simultaneous creation of ROS [127,128]. This hypothesis proposes that intracellular ROS are responsible for NETotic cell death in the following ways:The process entails the release of neutrophil elastase (ELANE) and myeloperoxidase (MPO) from the granules of neutrophils, subsequently leading to their relocation from the cytosol to the nucleus [129].The enhancement of ELANE’s MPO-dependent proteolytic activity [129].

Activated ELANE in the cytoplasm may accelerate F-actin proteolysis, reducing cytoskeleton dynamics. Furthermore, with regard to MPO, it is worth noting that the nuclear reservoir of elastase has the potential to initiate the degradation of histones and potentially even the nuclear envelope. As a result of this phenomenon, the chromatin fibers become intertwined with both cytoplasmic and nuclear constituents, leading to their extrusion and ultimately resulting in the rupture of the plasma membrane and regulated cell death [25].

Intracellular multiprotein complexes known as inflammasomes are responsible for the recruitment and activation of inflammatory caspases. Specifically, human procaspases 4 and 5, as well as murine procaspase-11, play crucial roles as constituents of the non-canonical inflammasome, which is activated in response to the intracellular presence of lipopolysaccharides (LPS) from specific Gram-negative bacterial pathogens [130]. In contrast, procaspase-1 functions as the primary effector protease in canonical inflammasomes. The canonical inflammasome sensor proteins, including NLRP1, NLRP3, NLRC4, AIM2, and pyrin, have been extensively studied in macrophages, and their involvement in the identification of PAMPs and DAMPs has been well established [131]. The canonical inflammasome sensors exhibit responsiveness to a wide range of PAMPs, including microbial nucleic acids, bacterial secretion systems, and components of microbial cell walls. Additionally, they are activated by environmental stressors and endogenous DAMPs that indicate harm to host cells during sterile inflammation [132]. Examples of such DAMPs include uric acid crystals, elevated levels of extracellular adenosine triphosphate (ATP), and the presence of mitochondrial and nuclear nucleic acids in the cytosolic compartment. Upon activation, caspase-1 enzymatically cleaves the pro-inflammatory proteins interleukin (IL)-1β and IL-18, resulting in the production of bioactive cytokines. Concurrently, inflammatory caspases facilitate the enzymatic cleavage of the pore-forming protein GSDMD. This process leads to the release of the N-terminal domain of GSDMD, which subsequently assembles into multimers within the plasma membrane [130,131,132]. These multimers induce the formation of sizable GSDMD pores, causing the permeabilization of the plasma membrane and resulting in the pyroptotic cell lysis of activated macrophages. The induction of canonical inflammasomes in macrophages facilitates pyroptotic cell lysis. Additionally, it has been proposed that the cleavage of GSDMD, which is mediated by neutrophil elastase, may increase the development of NETs produced by phorbol 12-myristate 13-acetate (PMA), independent of inflammasome activation [133].

### 2.5. Ferroptosis: Iron-Dependent Cell Death

Different from other kinds of cell death such as apoptosis and necrosis, ferroptosis is a recently discovered iron-dependent cell death. Iron (Fe) is the fourth most prevalent element in the Earth’s crust and is quite important for biological processes [134]. Fe acts as a co-factor in protein functions involved in the tricarboxylic acid (TCA) cycle and the electron transport chain, and it contributes to oxygen transport, DNA biosynthesis, and ATP production, all of which are required for cell viability [135]. Furthermore, iron has been demonstrated to be linked to the development and metastasis of tumors, suggesting that iron metabolism abnormalities may promote tumor expansion [136,137]. Fatty acid lipid peroxidation is dramatically sped up in the presence of iron, especially divalent iron. ROS are byproducts of the iron-dependent oxidative phosphorylation that occurs in the mitochondria during energy production [137]. When ROS levels surpass the cell’s anti-oxidation capacity, an oxidative stress response is triggered, which may cause harm or death to the cell via direct and indirect damage to big molecular components including proteins, nucleic acids, and lipids [138]. The program triggers a cascade of events beginning with thiol metabolism, continuing with lipid metabolism, and ending with iron-dependent lipid peroxidation and cell death [139]. Ferroptosis is a form of RCD that is induced by disruptions in the intracellular environment caused by oxidative stress. This process is constantly regulated by the enzyme glutathione peroxidase 4 (GPX4) and can be prevented by the use of iron chelators and lipophilic antioxidants [140]. Ferroptosis, which develops from an increase in iron-dependent lipid peroxide, is distinct from apoptosis and necrosis in the conventional sense [141]. Acute lipid peroxidation is a key initiator of ferroptosis, which also requires the presence of ROS and iron. As some chemical pathways that trigger ferroptosis are clear now, we know that hazardous lipid peroxide accumulation is associated with ferroptosis-regulated cell death [142]. The occurrence of a necrotic morphotype is observed in the context of ferroptosis, a form of cell death characterized by various mitochondrial changes such as shrinkage, an electron-dense ultrastructure, diminished or absent cristae, and ruptured OMM. Importantly, this process is not reliant on caspases, necrosome components, or cyclophilin D (CYPD), and operates independent of the molecular machinery involved in autophagy [143,144]. In the context of ferroptosis, it has been seen that mitochondria undergo a reduction in size as a result of weakened cristae and the occurrence of collapsed and ruptured membranes. However, it is worth noting that the cell membrane often remains intact, and the nucleus maintains its normal size while lacking chromatin condensation [140,145].

Defects in system XC^−^ and GPX4 axis cause the failure of glutathione-dependent antioxidant defense. GSH production requires extracellular cystine, which is brought into the cell via system XC^-^ and converted to cysteine. Changes in the cell’s cytology, such as a decrease in cell volume and an increase in the density of the mitochondrial membrane, are hallmarks of ferroptosis cell death [146]. There are two types of small-molecular chemicals that may trigger ferroptosis cell death. As inducers of class I ferroptosis, DPI2, erastin, sulfasalazine (SAS), and buthionine sulfoximine may cause an oxidation–reduction imbalance by inhibiting the system XC^-^ and decreasing the intracellular glutathione concentration [135]. Direct inhibition of GPX4 by class Π ferroptosis inducers such as RAS-synthetic lethal 3 (RSL3) compounds, DPI7, DPI10, DPI12, DPI13, etc., might result in lipid peroxide buildup [147]. Further, certain pharmaceuticals such as Sorafenib, Artemisinin, and its derivatives have been shown to promote ferroptosis [135] (Figure 4).

Ferroptosis prevention is applied by two enzymatic systems in the antioxidant reaction: GPx4 catalyzes the reduction of lipid peroxides in a glutathione-dependent reaction, and the recently identified FSP1 catalyzes the ubiquinone regeneration (coenzyme Q10, CoQ10), which acts as a lipid peroxyl radical decoy. In the ferroptosis cell death mechanism, there is no expansion of cytoplasm and organelles or rupture of the plasma membrane [148]. Furthermore, in contrast to pyroptotic cells, ferroptotic cells do not exhibit blebbing or the loss of plasma membrane integrity. Mitochondria seem smaller than usual with an increased membrane density, and this is the only morphological property of ferroptosis. During ferroptosis, mitochondria shrink, membranes break, ROS are released, iron overload occurs, and intracellular GSH is depleted [25,149].

### 2.6. Autophagy

Damaged organelles, misfolded proteins created during biosynthesis, and nonfunctional, prolonged-live proteins are recycled by lysosomes with a process called autophagy [150,151]. Micro-autophagy, chaperone-mediated autophagy (CMA), and macro-autophagy are the three main categories of autophagy that have been identified based on the process by which intracellular components are transported to the lysosome for destruction [152,153]. Prolonged-live proteins, damaged organelles, and defective proteins produced during biosynthesis are all cleared out of the cell by the self-digesting process known as autophagy. It has been shown that the autophagic process is supposed to regulate many distinct cellular activities, including proliferation, differentiation, adaptation to nutritional deprivation and oxidative stress, apoptosis, and the recycling of macromolecules and organelles [58].

Micro-autophagy entails the direct sequestration of cytoplasmic components into the lysosomes with acidic hydrolase degradation. It has been shown that proteins carrying the KFERQ motif (Lys-Phe-Glu-Arg-Gln) are specifically targeted by CMA. In this process, chaperones identify target proteins, which are then to be joined by lysosomes and destroyed [153]. When cells engage in autophagy, they create a double-membraned vesicle that merges with lysosomes to degrade cytoplasm, misfolded proteins, prolonged-lived proteins, and damaged organelles. There are 16 autophagy-related proteins involved in the intricate process of vesicle production (Atg proteins). Autophagy is accompanied by two ubiquitin-like conjugation systems. The development and size of the autophagosome are regulated by the complexes of autophagy regulators, Atg16-Atg12-Atg5, and Atg8-PE, that are produced by these systems [154]. Next, autophagosomes can be nucleated, expanded, uncovered, and joined with lysosomes after a complete construction. The formation of autophagosomes begins with the interaction of two complexes [155,156]:Class III PI3K Vps34, Beclin1/Atg-6, Vps15/p150.73, and Atg-14 form a complex that is referred to as the PI3K complex [157].The serine/threonine kinase Atg-1 [158].

Two more autophagy proteins, Atg-8/ Atg-13 and Atg-17, are required for the kinase activity of Atg-1. In mammals lacking Atg-13, Atg1 was shown to interact with Atg-8 orthologues such as microtubule-associated protein light chain 3 (LC3), G-amino butyric acid type A receptor-associated protein (GABARAP), and Golgi-associated ATPase enhancer of 16 KDa (GATE-16) [154].

To initiate the process of autophagosome formation, it is necessary for a reactive glycine residue within the soluble Atg-8 protein to become exposed through carboxyl-terminal cleavage mediated by the cysteine protease Atg-4. The activation of Atg-4 requires the participation of both the Atg-7 (E1-like) and Atg-3 (E2-like) enzymes [159,160]. The functionality of Atg-3 requires the presence of a protein complex comprising Atg-5, Atg-12, and Atg-16. Following the activation of Atg-8, a process occurs wherein phosphatidyl ethanolamine becomes covalently attached to the protein. In yeast, this modified protein is referred to as Atg-8-PE, while in humans, it is known as lipidated LC3-II. The segment in question persisted in its attachment to the autophagosome membrane until it was subjected to degradation by Atg-4 for the purpose of recycling. If Atg-8 remains covalently attached to the membrane without being cleaved by Atg-4, it has the potential to function as a signaling mechanism for autophagy [161]. Once the autophagosome has completed its formation, the complex consisting of Atg-16, Atg-5, and Atg-12 dissociates from the surrounding membrane. Subsequently, the components of this complex engage in a recycling process facilitated by Atg-2, Atg-18, and Atg-9. The formation of the autophagosome has reached its last stage, rendering it ready for a subsequent fusion with either an endosome or a lysosome [161].

Nucleophagy, a specific form of autophagy that selectively targets nuclear components for degradation, has been widely used as a model system to investigate selective macro-autophagy. Additionally, it has been essential in understanding the function of the core autophagic machinery in micro-autophagy. Nucleophagy has been observed as a mechanism that is implicated in several disease states, including cancer, neurodegeneration, and aging. Nucleophagic mechanisms are inherent to cellular growth and may also serve as a cellular response to diverse stress stimuli. The autophagic mechanism facilitates the transport of micronuclei, tiny pieces of nuclear material, to the vacuole for eventual destruction [162]. Evidence hypothesized that nucleophagy could be a mechanism to maintain nuclear and genome integrity in normal (noncancerous) cells, in response to DNA-damaging agents [163]. Nuclear abnormalities exhibit notable prominence in degenerative diseases and progeria disorders. The process of selective autophagy of organelles plays a crucial role in the maintenance of cellular homeostasis and the prevention of premature aging. Despite the nucleus playing a crucial role in cellular function by protecting our genetic material and regulating gene expression, our understanding of nuclear autophagy remains limited [164]. Nucleophagic mechanisms manifest under many situations. Several factors contribute to the increased occurrence of nucleophagy, including starvation, the inactivation of TORC1 caused by rapamycin, genotoxic stress, and the enlargement of the nucleus vacuole junction (NVJ), as well as abnormalities in the nuclear envelope and lamina [165]. In-depth examinations into the commencement of macro- and micronucleophagy have shown that these processes are contingent upon the Nem1/Spo7-Pah1 axis, which is a downstream mechanism resulting from the inactivation of TORC1 [166]. The Nem1/Spo7–Pah1 axis, a component of lipid metabolism, has a role in many autophagic mechanisms, such as endosomal sorting complexes needed for transport (ESCRT)-dependent micro-ER-phagy and autophagy-independent activities [167]. The occurrence of nucleophagy in yeast was first shown under conditions of nutritional constraint. The inhibition of TORC1 by nitrogen deprivation, as well as the use of rapamycin, causes nucleophagy regardless of the specific mechanism involved [168]. The nucleophagic processes are initiated by the creation of micronuclei. Inactivation of TORC1 triggers the recruitment of the essential autophagic machinery to these micronuclei [168,169]. Furthermore, the process of nucleophagy, which involves the degradation of nuclear components, is dependent on the presence of the cargo receptor Atg39 located in the nuclear envelope [168,170]. The presence of two stress-response element (STRE) repeats in the upstream region of ATG39, together with its upregulation of gene expression under stationary phase and nutritional deprivation circumstances, suggests a potential association with nucleophagic activity in response to starvation [171]. The presence of Nvj1, a protein located in the outer nuclear membrane (ONM), is crucial for the development of the nuclear-vacuolar junction (NVJ) and, therefore, the process of micronucleophagy. The upstream region of NVJ1 is reported to have two STRE repeat regions, as documented in reference [162]. It is evident that the overexpression of Nvj1 results in the expansion of the nuclear-vacuolar junction (NVJ), which therefore leads to increased rates of micronucleophagy [172].

In addition to shared characteristics in the control of expression, the localization of both Atg39 and Nvj1 is contingent upon the Nem1/Spo7-Pah1 axis [166]. The activation of the Nem1/Spo7-Pah1 axis occurs as a result of TORC1 inactivation, and the lack of this axis results in the impairment of both macro- and micronucleophagy [166,173]. The Nem1/Spo7 complex has been previously shown to be necessary for the preservation of nuclear envelope shape. This complex is specifically localized inside the nuclear envelope [174]. Although the complex’s impact on nucleophagy is notably significant, it also has relevance for several other forms of autophagy. However, it does not exert any effect on the Cvt-pathway [166]. The enzyme Pah1 catalyzes the dephosphorylation of phosphatidic acids, resulting in the production of diacylglycerol. The ability to sustain the integrity of the nuclear envelope may potentially play a role in the modulation of the nuclear membrane for the generation of micronuclei by the Nem1/Spo7-Pah1 axis.

The destruction of vacuolar membranes, whether as a target or a side consequence of microautophagic activities, is a characteristic that sets this process apart. The functioning of both macro-autophagy and micro-nucleophagy, as well as the preservation of nuclear shape, necessitates the involvement of Pah1 in diacylglycerol production [166]. The significance of the Nem1/Spo7-Pah1 axis extends beyond nucleophagy, including the broader context of microautophagy. This axis plays a crucial role in both autophagy-independent micro-ER-phagy, which relies on core autophagy mechanisms, and ESCRT-dependent micro-ER-phagy [167]. The probable explanations for the preserved importance of lipid supply and/or metabolism following microautophagic membrane consumption and participation in vacuolar domain formation have been proposed [166,167,175]. The nucleophagic activity seen in mammalian cells has been shown to be associated with both oncogenic and genotoxic stress [176]. The initiation of nucleophagy in mammalian cells occurs as a result of pathogenic circumstances. However, it is worth noting that the Nem1/Spo7–Pah1 axis, which is responsible for this process, is conserved across different organisms ranging from yeast to mammalian cells. Specifically, the orthologous CTDNEP1/NEP1R1-lipin complex serves a similar function in this regard. The CTDNEP1/NEP1R1-lipin complex is situated in the nuclear envelope, similar to its corresponding entity [177]. The functional complementation of yeast Nem1 by human CTDNEP1 has been shown, and mutations in CTDNEP1 have been identified in tumor cells [177,178]. The investigation of nucleophagy in mammalian cells under physiological settings remains a topic that requires more elucidation in future research endeavors.

### 2.7. Entosis

There are several mechanisms by which cell death may be triggered when a cell is engulfed by another cell, creating a “cell-in-cell” arrangement. During carcinogenesis and development, entosis is one mechanism that causes cell-in-cell formation [179]. Entotic cells invade their hosts and actively assist in their own engulfment; they are subsequently destroyed in a non-cell-autonomous manner. Detachment of the ECM by integrins is thought to be the initiating event in ectoderm neogenesis [180]. In contrast to phagocytosis, which does not rely on the existence of epithelial adherens junctions, the process of engulfment through entosis does necessitate the presence of these junctions. These junctions consist of the cell–cell adhesion receptor E-cadherin and the adherens junction/cytoskeleton linker protein-catenin [181]. Entotic cells, as opposed to phagocytized cells, engage in the active regulation of their own uptake by means of the RhoA GTPase activity and the RhoA effector kinases, such as Rho-associated, coiled-coil-containing protein kinase (ROCK) I/II [182]. Previous studies have demonstrated that the upregulation of RhoA or ROCK I/II is capable of inducing the internalization of cells expressing epithelial cadherin. This suggests that entosis has a greater resemblance to a cellular invasion process, resulting in the formation of cell-in-cell structures, rather than being a spontaneous engulfment phenomenon [183,184]. In the process of entotic cell death, the subsequent engulfment is succeeded by lysosome-mediated breakdown, which exhibits distinct characteristics compared to autophagy. The autophagic protein LC3 does not participate in the process of autophagosome formation [185]. Instead, LC3 is targeted by Vps34, ATG-7, and ATG-5 for achieving lipidation in the single membrane vacuole harboring the engulfed cell, where it promotes lysosome fusion and lysosome-mediated destruction [186,187]. Notably, not all invading cells die in the lysosome after undergoing the entosis process. Thus, entotic cells may survive within host cells and even grow if they manage to escape [179].

### 2.8. Methuosis

Methuosis, a non-apoptotic cell death, is a cellular process that is distinguished by the hyperactivation of Ras, and the significant accumulation of large vacuoles surrounded by a singular membrane. The vacuoles in question are derived from macro-pinosomes [188]. Interestingly, the involvement of the PI3K signaling pathway and the classic Ras-Raf-MEK-Erk axis is not observed in methuosis. The morphological characteristics of methuosis closely resemble necrosis, as a result of the cellular expansion and the disintegration of the plasma membrane. Researchers frequently employ electron microscopy to assess the distinctive view of methuosis [189]. The resulting Ras activation in methuosis increases micropinocytosis by activating a member of the Rac-1 protein family. Concurrently, the absence of Arf6-GTP hinders the process of macro-pinosome recycling [190]. The cytoplasm undergoes significant vacuolization as a consequence of the abnormal merging of newly formed macro-pinosomes. During the initial stages of methuosis, vacuoles are decorated with late endosomal markers, including lysosome-associated membrane protein 1 (LAMP1) and Rab7 [191]. Finally, cell death results from the accumulation of large vacuoles that cannot be recycled or fused with lysosomes. Vacuolization pathways are mediated by methuosis stimulants; there are two classes of methuosis inducers that trigger methuosis cell death:Class I is activated by Ras oncogenes, which induce vacuole formation by multiple sequential processes. The activation of Rac1 induces the process of macro-pinocytosis. Furthermore, the activated version of Rac1 interacts with G-protein-coupled receptor kinase-interacting protein 1 (GIT-1) to deactivate ADP-ribosylation factor 6 (Arf-6), thereby impeding the recycling of macro-pinosomes back to the plasma membrane. Consequently, the accumulating macro-pinosomes exhibit certain characteristics of late endosomes and subsequently merge together to form vacuoles [192].Vacuole development in class II methuosis inducer including mitogen-activated protein kinase kinase 4 (MKK-4), casein kinase 1 (CK1), nucleolin (Nuc), Arf6, GIT1, nerve growth factor (NGF), and early endosome antigen 1 (EEA1) [192].

### 2.9. Paraptosis

Paraptosis is characterized by the extensive vacuolization of the cytoplasm from either the enlarged endoplasmic reticulum (ER) or the mitochondria. Several studies have linked paraptosis to the osmotic expansion of the ER lumen and mitochondria for vacuolization due to the production of ROS and the accumulation of misfolded proteins in the ER [193]. Currently, there is a dearth of specific diagnostic assays for the identification of paraptosis, similar to the existing limitations in detecting entosis and methuosis. This situation is characterized by the presence of many cytoplasmic vacuoles with a single membrane when observed under an electron microscope. Although commonly regarded as a pro-survival regulator, there is evidence suggesting that the activation of the insulin-like growth factor 1 receptor (IGF1R) and its subsequent signaling pathways, including as MAPKs and JNK pathways, could potentially induce paraptosis [194].

### 2.10. Mitoptosis

Mitoptosis, alternatively referred to as mitochondrial suicide, is a different process from mitophagy, which includes the autophagic degradation of mitochondria. Mitoptosis operates by the modulation of mitochondrial dynamics, specifically controlled fission and fusion, with the aim of compromising their capacity to generate ATP [2]. Therefore, mitoptosis may be connected to both apoptosis and autophagy [195,196]. Mitochondria that have undergone destruction are engulfed by autophagosomes or converted into mitoptotic entities, afterwards being expelled from the cellular environment. In the present context, the term “Mitoptosis” pertains to a process of mitochondrial demise, as opposed to a mechanism of cellular death [197]. However, an excessive fission of mitochondria leads to their severe fragmentation and, in the end, cell death. Mechanistically, when BAX/BAK cause mitochondrial outer membrane permeabilization (MOMP), a protein called translocase of inner mitochondrial membrane 8a (TIMM-8a/DDP) is released from the mitochondrial intermembrane gap [198]. Later, DDP is translocated to the cytoplasm and binds to Dynamin Related Protein 1 (DRP1). Mitoptosis and mitochondrial fission are triggered by DDP’s interaction with DRP1, which results in DRP1’s recruitment and retention in the mitochondria. However, after much study, the process’s physical characteristics remain the primary means of description [199].

Mitophagy and mitochondrial dynamics are related. Fusion of OMMs is mediated by the small GTPases Mitofusin 1 (Mfn-1) and Mitofusin 2 (Mfn-2), whereas the fusion of IMMs is mediated by the large GTPase optic atrophy 1 (OPA-1) [200]. The shift of the GTPase Drp1 is essential for mitochondrial fission. DRP-1 binds to four different receptors—fission 1 (Fis1), mitochondrial fission factor (Mff), mitochondrial dynamics protein of Mid49, and Mid51—to anchor itself to the OMM [201]. Drp1’s location in the cytoplasm or on the OMM is influenced by post-translational modifications, including phosphorylation. It has been shown that Drp1 may be phosphorylated by sirtuin (SIRT), extracellular signal-regulated kinase (ERK), protein kinase A (PKA), P38/mitogen-activated protein kinase (MAPK), and AMP-activated protein kinase (AMPK) [202]. Mitochondrial fission is suppressed by the phosphorylation of Drp1 at Ser637 and Ser656 [203] but is stimulated by phosphorylation at Ser616, Ser579, and Ser600 [204,205,206]. DRP-1 activity is downregulated by PKA-mediated phosphorylation at serine 656, leading to hyperfused mitochondrial networks, and reactivated by the phosphatase calcineurin. Mitochondria are shrunken and damaged when DRP-1 is active [207]. When it comes to quality control measures before mitophagy, DRP-1-driven mitochondrial fragmentation cannot be overlooked [2,202].

### 2.11. Parthanatos

Parthanatos is characterized by the hyperactivation of Poly(ADP-ribose)polymerase (PARP) and is a kind of mitochondria-linked, caspase-independent cell death [208]. When Poly(ADP-ribose) (PAR) is synthesized by PARP, it is then shuttled from the nucleus to the cytoplasm [209], where it interacts with mitochondrial proteins, and apoptosis-inducing factor (AIF) is released [210,211]. Transporting of free AIF from mitochondria to the nucleus triggers chromatin condensation and DNA damage. The apoptotic process requires PARP cleavage, while the parthanatos process requires intact PARP and PARP activation. Further, parthanatos is independent of caspases since it is resistant to the inhibition by broad-spectrum caspase inhibitors [212]. Apoptotic bodies are not produced during parthanatos. Moreover, unlike the modest DNA fragmentation generally seen in apoptosis, the DNA fragmentation in the parthanatos is on a much larger scale. To a practical extent, parthanatos biomarkers include PARP-1 activity, PAR accumulation, and nuclear AIF. Depolarization of mitochondria, as measured using fluorescent probe labeling, provides more evidence on the process [213].

### 2.12. Necroptosis

Necroptosis, also called programmed necrosis [15], is switched on by the activation of receptor-interacting protein kinases (RIPKs) in an interaction with cell surface receptors by several signaling pathways, i.e., the T-cell receptor, TLRs, and DRs are specifically engaged in the RIPKs’ activation [214]. The main components of necrosome include RIPK-1 and RIPK-3 [215]. The RIPK-3 protein exerts its functional role by facilitating the activation of the downstream molecule known as mixed lineage kinase domain-like protein (MLKL) through the process of phosphorylation. This phosphorylation event subsequently triggers the oligomerization of MLKL [216]. Oligomerized MLKL penetrates and permeates the cell membrane, ultimately leading to cell death [217]. In addition, the presence of a viral infection or the presence of double-stranded viral DNA triggers the activation of the cytosolic DNA sensor. Then, this sensor, known as DNA-dependent activator of interferon (DAI) regulatory factors, initiates the process of RIP3-dependent necroptosis [218]. Through necroptosis, the necrotic morphology of shattered membranes and missing organelles is manifested. Necroptosis can be evaluated in a number of ways, including the use of cell-impermeable DNA binding dyes to detect a breakdown of the plasma membrane, Western blotting for the release of proteins such as fluorescent probes to examine mitochondrial potential, high mobility group box 1 (HMGB1), lactate dehydrogenase LDH, and Cyclophilin-A, and electron microscopy to examine morphology [2,219,220]. Alternative proposed approaches include the use of specific necroptosis inhibitors, such as Necrostatin-1 [221], and the measurement of key proteins in the pathway [222,223].

### 2.13. Necrosis

Necrosis, the Greek term that means “to kill”, is the phrase often used to describe cell death [14,224]. Necrosis refers to the irreversible cell damage and subsequent cell death caused by pathogenic processes [93]. This particular form of cellular demise involves the inflation of organelles, the rupture of the plasma membrane, and the subsequent lysis of the cell, thereby leading to the release of its contents into the adjacent tissue and resulting in detrimental effects. Necrosis is a type of cellular demise that is typically followed by inflammatory reactions due to the liberation of many molecules, including DNA, ATP, heat shock proteins, uric acid, and nuclear proteins. These molecules stimulate the activation of inflammasomes and the subsequent release of the pro-inflammatory cytokine interleukin-1 beta (IL-1β). The rupture of the cell membrane in response to a painful stimulus facilitates the influx of extracellular ions into the cell, subsequently leading to the entry of fluid. This influx of ions and fluid results in the swelling of the cell and its organelles, a phenomenon referred to as necrosis [225].

Proteolytic enzymes, such as proteases, RNAases, DNAases, and phosphatases, are released into the cell once the lysosomal membrane is disrupted. Damage to DNA, RNA, and proteins results from their activation in the cytosol [226]. These enzymes break down cells by digesting their constituent parts. Disruption of the plasma membrane, allowing intracellular contents to leak out into the surrounding tissue, is a common result of both of these mechanisms. Necrotic cells are characterized morphologically by the enlargement of cellular organelles such as the endoplasmic reticulum and mitochondria, plasma membrane rupture, and eventual cell lysis [97]. Eosinophilic, glassy, and vacuolated cellular changes result from these alterations. The breakdown of cell membranes and organelle membranes contributes to necrosis. The first biochemical alteration noticed after injury is a reduction in ATP production or depletion. In the presence of oxygen, ATP is synthesized by a process called oxidative phosphorylation in the mitochondria [226,227]. A lack of oxygen delivery to cells causes necrosis in response to hypoxia or chemical insult, which in turn reduces ATP generation [228].

Due to the inability to pump sodium out of the cell, cells expand and ribosomes detach from the endoplasmic reticulum because the energy-dependent sodium pump in the plasma membrane fails. Damage to the mitochondria is caused by both the elevated levels of calcium in the cytosol and the oxidative stress. Calcium in the cytosol activates a number of cytosolic enzymes, including phospholipases and proteases, which in turn break down membranes (particularly lysosomal membranes) and proteins [229,230]. Inflammation is often seen in cases of necrotic death [1]. HMGB1 and hepatoma-derived growth factor (HDGF) are two of the factors released by necrotic cells. The nod-like receptor protein 3 (NLRP3), the main protein of inflammasome, is responsible for sensing HMGB1 and HDGF. Consequently, this triggers inflammasome activation, which in turn releases the pro-inflammatory cytokine IL-1β. When cells are injured, they release ATP, which is then used to activate the NLRP3 inflammasome [97].

## 3. Conclusions

As extensively examined before, the cell death plays a significant role in several biological processes, including development, maintenance of tissue equilibrium, inflammatory responses, immune system function, and several pathological states. On the one hand, cell death is a significant causal factor in illnesses characterized by the permanent loss of post-mitotic tissues, such as myocardial infarction and dementia. Conversely, dysfunctions in the molecular signaling cascades that initiate regulated cell death are linked to pathological conditions defined by abnormal cellular proliferation or accumulation, such as some autoimmune illnesses and cancer. Therefore, the targeting of cell death emerges as a prominent therapeutic strategy for the treatment of many human medical conditions. Molecular mechanisms of cell death could provide new insights into the cell cycle processes and potential molecular components to prevent and stimulate cell death in different signaling pathways and diseases. In summary, the targeting of cell death mechanisms presents a promising avenue for treating numerous human disorders. In addition to the potential concerns associated with the pharmacokinetics and pharmacodynamics of the compounds that have been examined thus far, the intricate interconnectedness of the signaling modules responsible for regulating regulated cell death in mammalian organisms remains a concern. Therefore, while it may seem relatively straightforward to favor cell death in special conditions, inhibiting it becomes challenging after crossing a previously undefined threshold. This may need the simultaneous inhibition of many signal transduction modules, making it a difficult goal to achieve. Nonetheless, further studies are necessary to develop the most effective strategies for utilizing cell death modulators in a clinical context.

## Figures and Tables

**Figure 1 biology-12-01426-f001:**
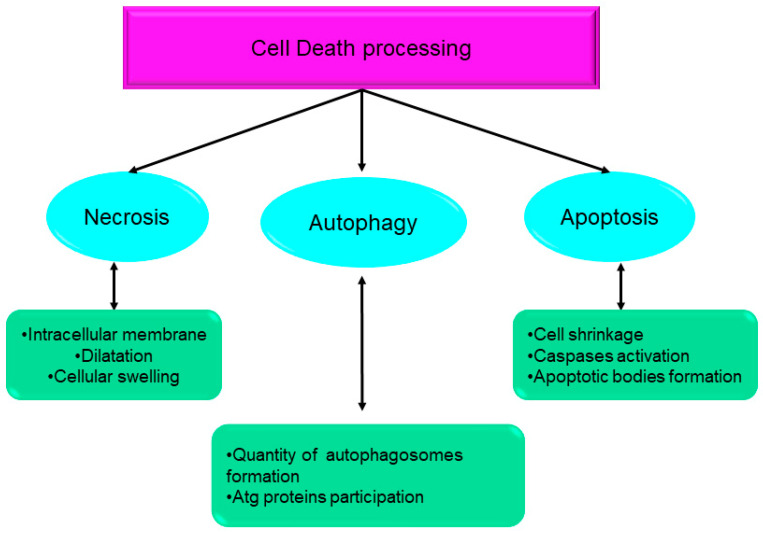
Systematic classification and analysis of cell death, encompassing morphological, biochemical, and functional aspects. Atg: autophagy-related gene.

**Figure 2 biology-12-01426-f002:**
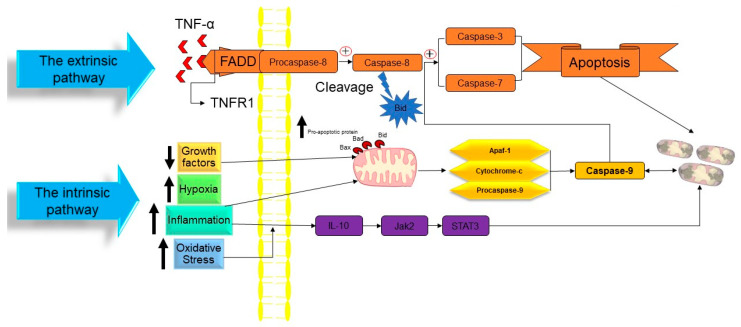
The molecular mechanism of apoptotic cell death. TNF: tumor necrosis factor, TNFR1: tumor necrosis factor receptor 1, FADD: FAS-associated death domain, Bid: BH3 interacting domain death agonist, Bad: BCL2-associated agonist of cell death, Bax: Bcl-2-associated X protein, Apaf1: apoptotic protease activating factor-1, IL-10: interleukin 10, Jak2: janus kinase 2, and STAT3: signal transducer and activator of transcription 3. ↑ Indicated increasing of the factor. ↓ Indicated reducing of the factor.

**Figure 3 biology-12-01426-f003:**
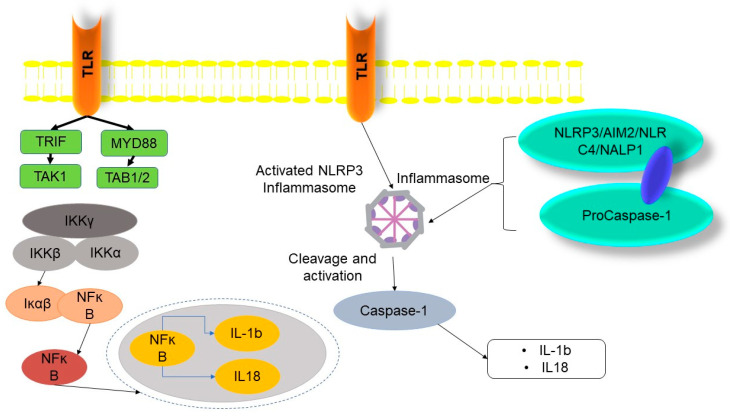
Molecular mechanism of pyroptosis in the form of lytic programmed cell death is characterized by its highly inflammatory nature and is most frequently observed following intracellular pathogen infections. TLR: Toll-like receptor, TRIF: TIR-domain-containing adapter-inducing interferon-β, MYD88: myeloid differentiation primary response protein 88, TAk1: transforming growth factor (TGF)-β-activated kinase 1, TAB: TAK1-binding protein, IKK: nuclear factor kappa-B kinase, NF-κB: nuclear factor kappa B, NLRP3: nod-like receptor family pyrin domain containing 3, AIM2: absent in melanoma-2, NLRC4: nod-like receptor family CARD domain containing 4, and NALP1: nucleotide-binding oligomerization domain-like receptor family, pyrin domain-containing 1. Purple ellipse indicted a complex.

**Figure 4 biology-12-01426-f004:**
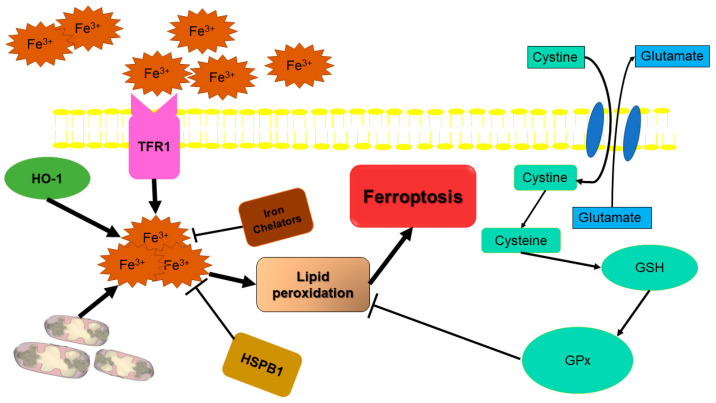
Molecular mechanism of ferroptosis as an iron-dependent cell death. Fe^3+^: ferric ion, TFR1: transferrin receptor 1, HSPB1: heat shock protein beta-1, GPX: glutathione peroxidase, and GSH: glutathione. The blue ellipse is the channel.

**Table 1 biology-12-01426-t001:** Difference between anoikis and apoptotic cell death based on the definition, induction, and function.

Anoikis	Apoptosis
Programmed cell death, occurring in cells, separated from the extracellular matrix. Induced upon the separation of a cell from the extracellular matrix. Prevents the adherent-independent cell growth and the attachment of cells to an improper matrix, thus preventing the colonization of distant organs.	Programmed cell death, occurring in cells that are redundant, functionally incomplete, or dangerous for an organism. Induced when a cell becomes redundant, functionally incomplete, or dangerous for an organism. Mostly removes useful cells during the fethal/larval development and also the potentially harmful cells.

**Table 2 biology-12-01426-t002:** The difference between pyroptosis and apoptotic cell death based on morphological changes, molecular mechanism, regulation, and mitochondrial participation.

	Pyroptosis	Apoptosis
**Morphological Changes**	Cell shrinkage. Maintenance of the integrity of the plasma membrane. Formation of membrane blebs. Degradation of genomic DNA.	Cellular enlargement and subsequent rupture. Breakdown of the plasma membrane. Breakage of DNA. Generation of pores in the plasma membrane. Maintenance preservation of nuclear integrity. Occurrence of nuclear condensation.
**Molecular Mechanism**	Absence of an inflammatory reaction. Generation of apoptosome and liberation of Cytochrome-C. Activation of Caspases 8, 9, 3, 6, and 7. Activation of pro-apoptotic BCL-2 and DISC proteins.	Inflammatory cell death. Development of an inflammasome and activation of gasdermins. Activation of interleukin-1β (IL-1β) and interleukin-18 (IL-18). Activation of GSDME and NLRP3 inflammasome, via canonical and non-canonical routes.
**Regulation**	The apoptotic cell death occurs when the ubiquitination of RIPK1 is inhibited, leading to the formation of a complex between RIPK1, FADD, and pro-caspase-8. This complex activates caspase-8, which then cleaves RIPK1, ultimately resulting in the apoptotic cell death.	NLRP3 and other proteins are regulated by post-translational phosphorylation and ubiquitylation modifications.
**Mitochondrial Participation**	The release of Cytochrome-C from mitochondria and the subsequent generation of apoptotic bodies.	Mitochondria are engaged in the control of gasdermin D oligomerization and the consequent development of pores in the plasma membrane.

## Data Availability

Not applicable.

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
