# Peer review of "Types of Cell Death from a Molecular Perspective"

_biology, 2023, doi:10.3390/biology12111426_

Round 1
Reviewer 1 Report
Comments and Suggestions for Authors
In “Types of Cell Death from a Molecular Perspective”, the authors provide an overview of different cell death mechanisms. Molecular pathways and specific morphological features associated with different cell death mechanisms are described. The authors also highlighted that targeting cell death mechanisms could be promising for development of treatment strategies of various diseases.
However, the conclusion is very short for the review paper.
Here are my suggestions. Authors could also include a short description of strategies used to understand cell death mechanisms. For example, over last 20-25 years different computational models were developed to understand different cell death mechanisms, which allows one to study different treatment strategies in silico. For a recent review describing the different cell death models see the relevant to this review work: PMID: 37420422, doi: 10.3390/e24101402.
Also, different mechanisms may involve common molecular components. In conclusion, the authors could discuss similarities among different mechanisms. Some similarities are also highlighted in example work above.
This will make the review more valuable as the authors not only describe each mechanism but will provide a bigger picture on cell death regulation.
Comments on the Quality of English LanguageEnglish is fine
Author Response
Reviewer 1
Comments and Suggestions for Authors
In “Types of Cell Death from a Molecular Perspective”, the authors provide an overview of different cell death mechanisms. Molecular pathways and specific morphological features associated with different cell death mechanisms are described. The authors also highlighted that targeting cell death mechanisms could be promising for development of treatment strategies of various diseases.
Query1. However, the conclusion is very short for the review paper.
Response: Thanks a lot for the reviewer's attention. Based on your comment, we expanded and rewrote the conclusion section.
Query2. Here are my suggestions. Authors could also include a short description of strategies used to understand cell death mechanisms. For example, over last 20-25 years different computational models were developed to understand different cell death mechanisms, which allows one to study different treatment strategies in silico. For a recent review describing the different cell death models see the relevant to this review work: PMID: 37420422, doi: 10.3390/e24101402.
Response: Thanks a lot for the reviewer's attention. Based on your comment, we mentioned computational models in the introduction section to understand different cell death mechanisms, which allows one to study different treatment strategies in-silico.
Query3. Also, different mechanisms may involve common molecular components. In conclusion, the authors could discuss similarities among different mechanisms. Some similarities are also highlighted in example work above. This will make the review more valuable as the authors not only describe each mechanism but will provide a bigger picture on cell death regulation.
Response: Thank you for your recommendation. Based on your comment, we pointed to different mechanisms that may involve common molecular components in similar cell death.
Reviewer 2 Report
Comments and Suggestions for Authors
Dear authors,
The review paper “Types of Cell Death from a Molecular Perspective“ is in the scope of the special issue (Cell Self-Destruction (Programmed Cell Death), Immunonutrition and Metabolism) of the journal Biology. In focus of the paper are different pathways through which a cell can undergo cell death. Manuscript is clear, relevant for the field and presented in a well-structured manner. Cited references are mostly recent publications and relevant to the field. Statements are drawn coherent and supported by the listed citations. The pictures show the (cell death) mechanisms correctly, but in order to make them easier to follow and understand, it would be good to explain the abbreviations used, in the description of the figures. English language is appropriate and understandable. I suggest this paper to be published in Biology after suitable revision.
Below I cite some minor comments:
- species names should be written in italics (e.g. ln 57, 58 Dictyostelium discoideum, Escherichia coli)
- introduce explanation for the abbreviations (e.g. RCD, IAPs, APAF1, MPT, DR, DISC, TRADD, FADD, ECM, OMM, TNFR, HMBG1 RIPK, PARP, CMA ….) at first mention
- explain the abbreviations used in the figures in figure description
The manuscript should be revised before final acceptance for publication.
Author Response
Reviewer 2
Comments and Suggestions for Authors
Dear authors,
The review paper “Types of Cell Death from a Molecular Perspective “is in the scope of the special issue (Cell Self-Destruction (Programmed Cell Death), Immunonutrition and Metabolism) of the journal Biology. In focus of the paper are different pathways through which a cell can undergo cell death. Manuscript is clear, relevant for the field and presented in a well-structured manner. Cited references are mostly recent publications and relevant to the field. Statements are drawn coherent and supported by the listed citations. The pictures show the (cell death) mechanisms correctly, but in order to make them easier to follow and understand, it would be good to explain the abbreviations used, in the description of the figures. English language is appropriate and understandable. I suggest this paper to be published in Biology after suitable revision.
Below I cite some minor comments:
Query1. Species names should be written in italics (e.g. ln 57, 58 Dictyostelium discoideum, Escherichia coli)
Response: Thanks a lot for the reviewer's attention. Based on your comment, the format of species names was corrected.
Query2. introduce explanation for the abbreviations (e.g. RCD, IAPs, APAF1, MPT, DR, DISC, TRADD, FADD, ECM, OMM, TNFR, HMBG1 RIPK, PARP, CMA ….) at first mention
Response: Thanks a lot for the reviewer's attention. Based on your comment, we provided an introductory explanation for the abbreviations at first mention.
Query3. explain the abbreviations used in the figures in the figure description.
Response: Thanks a lot for the reviewer's attention. We explained the abbreviations used in the figures in the figure description.
Reviewer 3 Report
Comments and Suggestions for Authors
The presented review aims to provide an overview of various forms of cell death with molecular angles recently described in the literature. However, I would like to point out few notes and comments that might help the authors to better explain some concepts, in my opinion.
1. The acronymous RCD is never explained, I guess the authors meant "Regulated Cell Death", please add the extended proper nomenclature at page 2.
2. E. coli is not unicellular eukaryote, as reported here.
3. Authors describe DAMPs as endogenous warning signals without including PAMPs (the exogenous dangers) which are also a fundamental part in the RCD processes, please add it.
4. At page 3 authors report that caspase-independent cell death induces the breakage of the plasma membrane in non-apoptotic cell death. This is partly incorrect since caspases trigger the activation of Gasdermin proteins whose N-terminus fragment create pores within the membrane which, upon accumulation, end up to plasma membrane rupture as well. Therefore the following sentence: "In contrast, caspase-independent cell death and membrane rupturing are considered to be non-apoptotic cell death signs" is not fully correct since caspase-dependent cell death (such as pyroptosis) can cause membrane rupturing too.
4. At page 5 authors report, correctly, that caspase-3 activation induces apoptosis. However, gasdermin E, a substrate of active caspase-3 induces pyroptosis, rather than apoptosis. This concept should be mentioned. It would be also helpful for the readers if authors could describe their envision on how caspase-3 activation discerns between apoptotic and pyroptosis pathway once activated. Are these 2 pathways supported in a cell/tissue dependent context? Are they mutually exclusive? Or one happens to be faster than the other one? Same concept not only for apoptosis/pyroptosis but also for apoptosis/anoikis (page 7).
5. The pyroptosis paragraph should be better described from molecular perspectives. For instance, the executors of pyroptosis, the gasdermin family proteins, are not mentioned at all. The mechanistic pathway that carry to their full activation should be included. Moreover, multiple inflammasomes/caspases are involved in the pyroptosis cell death, not only NLRP3/caspase-1, which, in my opinion, should be also included in the description.
6. Similar concern for the section 2.4. Here the authors point out the NETs are formed upon the engagement of TLRs. This is also valid for the pyroptosis form of cell death, where, the first ligand is principally recognized by TLRs receptors. Therefore TLRs should also added in the section 2.3. Finally, it should also been included the recent literature that combines the pyroptosis to netosis, reporting that these two pathways cross-react in order to the NETs to be released from neutrophils.
7. ROS are key players not only in Ferroptosis but also in pyroptosis (ROS are actually considered the major activator of pyroptosis, therefore if they are mentioned in the section 2.5, they should also been mentioned much earlier in the review. Again, if ROS are involved in multiple cell death pathways, how the cells are actually able to distinguish which death pathway to pursue?
8. Methuosis is nowadays considered to be a non-apoptotic cell death pathway.
9. The autophagy of the nucleus (nucleophagy) as a potential form of cell death might been mentioned as well.
10. Apoptotic cell death during crisis (the apoptosis mediated by telomeres shortening) might be included too, as it is very relevant in cancer onset.
11. The last paragraph of the section 2.13, Necrosis, describing the role of the NLRP3 inflammasome should be mentioned in the paragraph 2.3.
12. The name of the proteins should be written in their fully form (as first time mentioned).
Comments on the Quality of English LanguageThe quality of English could be improved.
Author Response
Reviewer 3
Comments and Suggestions for Authors
The presented review aims to provide an overview of various forms of cell death with molecular angles recently described in the literature. However, I would like to point out few notes and comments that might help the authors to better explain some concepts, in my opinion.
- The acronymous RCD is never explained, I guess the authors meant "Regulated Cell Death", please add the extended proper nomenclature at page 2.
Response: Thanks a lot for the reviewer's attention. Based on your comment, we provided an introductory explanation for the abbreviations at first mention.
- E. coli is not unicellular eukaryote, as reported here.
Response: Thanks a lot for your attention. we corrected this mistake.
- Authors describe DAMPs as endogenous warning signals without including PAMPs (the exogenous dangers) which are also a fundamental part in the RCD processes, please add it.
Response: Thank you for your recommendation. We added PAMPs in the warning signals part in the RCD processes.
- At page 3 authors report that caspase-independent cell death induces the breakage of the plasma membrane in non-apoptotic cell death. This is partly incorrect since caspases trigger the activation of Gasdermin proteins whose N-terminus fragment create pores within the membrane which, upon accumulation, end up to plasma membrane rupture as well. Therefore, the following sentence: "In contrast, caspase-independent cell death and membrane rupturing are considered to be non-apoptotic cell death signs" is not fully correct since caspase-dependent cell death (such as pyroptosis) can cause membrane rupturing too.
Response: Thanks for pointing this out. We have added your comments to this section as an exception to this classification.
- At page 5 authors report, correctly, that caspase-3 activation induces apoptosis. However, gasdermin E, a substrate of active caspase-3 induces pyroptosis, rather than apoptosis. This concept should be mentioned. It would be also helpful for the readers if authors could describe their envision on how caspase-3 activation discerns between apoptotic and pyroptosis pathway once activated. Are these 2 pathways supported in a cell/tissue dependent context? Are they mutually exclusive? Or one happens to be faster than the other one? Same concept not only for apoptosis/pyroptosis but also for apoptosis/anoikis (page 7).
Response: Thanks a lot for the reviewer's attention. Based on your comment, we provided a paragraph about cross-talk and the differences between the apoptotic/pyroptosis pathway and apoptosis/anoikis and Tables 1 and 2. Moreover, we added gasdermin E, a substrate of active caspase-3 that induces apoptosis.
- The pyroptosis paragraph should be better described from molecular perspectives. For instance, the executors of pyroptosis, the gasdermin family proteins, are not mentioned at all. The mechanistic pathway that carry to their full activation should be included. Moreover, multiple inflammasomes/caspases are involved in the pyroptosis cell death, not only NLRP3/caspase-1, which, in my opinion, should be also included in the description.
Response: Thanks a lot for the reviewer's attention. We described molecular perspectives of pyroptosis, gasdermin family proteins, and inflammasomes/caspases that are involved in pyroptosis cell death.
- Similar concern for the section 2.4. Here the authors point out the NETs are formed upon the engagement of TLRs. This is also valid for the pyroptosis form of cell death, where, the first ligand is principally recognized by TLRs receptors. Therefore, TLRs should also added in the section 2.3. Finally, it should also been included the recent literature that combines the pyroptosis to netosis, reporting that these two pathways cross-react in order to the NETs to be released from neutrophils.
Response: We pointed out the engagement of TLRs in the pyroptosis in section 2.3. Moreover, we provided recent literature that combines pyroptosis with netosis, reporting that these two pathways cross-react in order for the NETs to be released from neutrophils.
- ROS are key players not only in Ferroptosis but also in pyroptosis (ROS are actually considered the major activator of pyroptosis, therefore if they are mentioned in the section 2.5, they should also been mentioned much earlier in the review. Again, if ROS are involved in multiple cell death pathways, how the cells are actually able to distinguish which death pathway to pursue?
Response: Thank you for your recommendation. We pointed out ROS as a major activator of pyroptosis and the mechanism of ROS in the activation of cell death processes.
- Methuosis is nowadays considered to be a non-apoptotic cell death pathway.
Response: Sorry for this mistake. We edited.
- The autophagy of the nucleus (nucleophagy) as a potential form of cell death might been mentioned as well.
Response: Thank you for your recommendation. We mentioned autophagy of the nucleus (nucleophagy) as a potential form of cell death in the Autophagy section.
- Apoptotic cell death during crisis (the apoptosis mediated by telomeres shortening) might be included too, as it is very relevant in cancer onset.
Response: We mentioned telomeres shortening as a relevant mechanism in Apoptotic cell death during a crisis.
- The last paragraph of the section 2.13, Necrosis, describing the role of the NLRP3 inflammasome should be mentioned in the paragraph 2.3.
Response: We mentioned the role of the NLRP3 inflammasome in the paragraph 2.3
- The name of the proteins should be written in their fully form (as first time mentioned).
Response: Thanks a lot for the reviewer's attention. Based on your comment, we provided an introductory explanation for the abbreviations at first mention.
Round 2
Reviewer 1 Report
Comments and Suggestions for Authors
Authors have addressed all my comments, the Review was significantly improved. My recommendation is to accept the manuscript for publication.
Comments on the Quality of English LanguageEnglish is good
Author Response
Thank you so much for your valuable comments.

Reviewer 3 Report
Comments and Suggestions for Authors
The authors have adequately responded to my comments
Author Response
Thank you so much.
